# Cytotoxic Effect Induced by Sicilian Oregano Essential Oil in Human Breast Cancer Cells

**DOI:** 10.3390/cells12232733

**Published:** 2023-11-29

**Authors:** Diana Di Liberto, Nicolò Iacuzzi, Giovanni Pratelli, Antonella Porrello, Antonella Maggio, Salvatore La Bella, Anna De Blasio, Antonietta Notaro, Antonella D’Anneo, Sonia Emanuele, Federica Affranchi, Michela Giuliano, Marianna Lauricella, Daniela Carlisi

**Affiliations:** 1Department of Biomedicine, Neurosciences and Advanced Diagnostics (BIND), Institute of Biochemistry, University of Palermo, Via del Vespro 129, 90127 Palermo, Italy; diana.diliberto@unipa.it (D.D.L.); sonia.emanuele@unipa.it (S.E.); 2Department of Agricultural, Food and Forest Sciences, University of Palermo, Viale delle Scienze, 90128 Palermo, Italy; nicolo.iacuzzi@unipa.it (N.I.); salvatore.labella@unipa.it (S.L.B.); 3Department of Physics and Chemistry (DiFC)-Emilio Segrè, University of Palermo, Via del Vespro 129, 90127 Palermo, Italy; giovanni.pratelli@unipa.it; 4Department of Biological, Chemical and Pharmaceutical Sciences and Technologies (STEBICEF), Section of Chemistry, University of Palermo, Viale delle Scienze, 90128 Palermo, Italy; antonella.porrello@unipa.it (A.P.); antonella.maggio@unipa.it (A.M.); 5Laboratory of Biochemistry, Department of Biological, Chemical and Pharmaceutical Sciences and Technologies (STEBICEF), University of Palermo, Via del Vespro 129, 90127 Palermo, Italy; anna.deblasio@unipa.it (A.D.B.); antonietta.notaro@unipa.it (A.N.); antonella.danneo@unipa.it (A.D.); federica.affranchi@unipa.it (F.A.); michela.giuliano@unipa.it (M.G.)

**Keywords:** breast cancer, essential oil, oregano, antioxidant activity, GC–MS

## Abstract

*Origanum vulgare* L. is an aromatic plant that exerts antibacterial, antioxidant, anti-inflammatory, and antitumor activities, mainly due to its essential oil (EO) content. In this study, we investigated the possible mechanism underlying the in vitro antitumor activity of EO extracted by hydrodistillation of dried flowers and leaves of *Origanum vulgare* L. grown in Sicily (Italy) in MDA-MB-231 and MCF-7 breast cancer cell lines. Gas chromatography–mass spectrometry analysis of Oregano essential oil (OEO) composition highlighted the presence of twenty-six major phytocompounds, such as *p*-cymene, *γ*-terpinene, and thymoquinone *p*-acetanisole. OEO possesses strong antioxidant capacity, as demonstrated by the DPPH test. Our studies provided evidence that OEO reduces the viability of both MCF-7 and MDA-MB-231 cells. The cytotoxic effect of OEO on breast cancer cells was partially counteracted by the addition of z-VAD-fmk, a general caspase inhibitor. Caspases and mitochondrial dysfunction appeared to be involved in the OEO-induced death mechanism. Western blotting analysis showed that OEO-induced activation of pro-caspases-9 and -3 and fragmentation of PARP decreased the levels of Bcl-2 and Bcl-xL while increasing those of Bax and VDAC. In addition, fluorescence microscopy and cytofluorimetric analysis showed that OEO induces a loss of mitochondrial membrane potential in both cell lines. Furthermore, we tested the effects of *p*-cymene, γ-terpinene, thymoquinone, and p-acetanisole, which are the main components of OEO. Our findings highlighted that the effect of OEO on MDA-MB-231 and MCF-7 cells appears to be mainly due to the combination of different constituents of OEO, providing evidence of the potential use of OEO for breast cancer treatment.

## 1. Introduction

The common oregano, scientific name *Origanum vulgare* L., is an aromatic perennial plant belonging to the *Lamiaceae* family widely used as a spice and as a medicinal plant. The species of this genus (about 40–50) prefer temperate-warm zones and originate mainly from the Mediterranean basin, although some species are also present in Asia [1,2].

From a botanical point of view, *Origanum vulgare* L. is a perennial plant with a woody base and herbaceous stems that grows up to a height of 20–80 cm. It is a very variable species, which includes several subspecies, such as, e.g., subsp. *hirtum* (Link) Ietsw., subsp. *glandulosum* (Desf.) Ietsw., subsp. *puny* (K. Koch), subsp. *virens* (Hoffmanns. & Link) Ietsw., and subsp. *viridulum* (Martrin-Donos) Nyman [3]. Oregano is used as a source of plant-derived drugs in modern herbal medicine thanks to its essential oils (EOs) [4,5,6,7]. EOs are synthesized in the glandular peltate trichomes, which are found on the surface of stems, leaves, and flowers (sepals, petals) [4]. Genetic and environmental factors influence the density and size of the glandular trichomes and, therefore, directly affect the essential oil yield [3,8]. Furthermore, the chemical composition of EO is influenced by the altitude of growth of the plant, the supply of water and nitrogen, the geographical area, and the harvest time and stage of maturity of the plant [9]. They contain monoterpenes and sesquiterpene hydrocarbons, as well as phenolic compounds [10]. Terpenes, such as thymol, carvacrol, *p*-cymene, *γ*-terpinene, and linalool, are the main constituents [10].

In the last few years, several studies have been carried out to evaluate the effect of biologically active derivatives obtained from EOs of medicinal plants on the development of new potential drugs for various pathologies with significant social impact [11,12,13], including cancer [5,14]. The EOs present in *Origanum vulgare* L. (OEO) are a valid source of biologically active compounds endowed with antibacterial, antifungal, antioxidant, anti-inflammatory, and antilipogenic properties [4,15,16,17,18,19,20].

In addition, an anticancer effect of OEO has been demonstrated on stomach cancer [16], breast adenocarcinoma [21], cervical adenocarcinoma [22,23], and colon adenocarcinoma [23] cell lines. 

Our study was designed to investigate the biochemical mechanism by which OEO obtained by Sicilian *Origanum vulgare* L. induces cell death in MDA-MB-231, a triple-negative breast cancer (TNBC) cell line, and MCF-7, a breast cancer (BC) cell line responsive to estrogen. We also evaluated OEO phytochemical composition using HPLC/MS (high-performance liquid chromatography/mass spectrometry) analysis. Our study demonstrated that OEO induced apoptosis in both breast cancer cell lines through the involvement of mitochondria and caspase activation. 

## 2. Materials and Methods

### 2.1. Plant Material

Essential oils were obtained from Sicilian *Origanum vulgare* ssp. *hirtum* (sin.: *O. heracleaticum* L.) collected in the country of Vallelunga (Sicily, Italy) at 637 m a.s.l. (named Biotype BIO 59). The plant material was collected from a collection field located at the “Orleans” experimental field of the University of Palermo (38°10′77.42″, 13°34′95.64″, 31 m a.s.l.). The collection field was made in 2017 from wild plants. Agamic propagation was accomplished by dividing the bushes. BIO 59 was collected in the country of Vallelunga (Sicily, Italy) at 637 m a.s.l.

### 2.2. Harvest of Plant Material

After being chopped at a height of 5 cm, the plants were allowed to dry for approximately 10 days at a temperature of 25–30 °C in a ventilated and shaded environment. Stems, leaves, and flowers were manually separated from the plant material. Only the dried flowering aerial parts of the plants—flowers and leaves—are used in EO extractions. 

### 2.3. Essential Oil Extraction

Following the standard procedure outlined in European Pharmacopoeia [24], 500 g of air-dried leaves and flowers were ground in a Waring blender and then subjected to hydrodistillation for 3 h or until increases in EO concentrations were no longer desired. In preparation for the GC and GC–MS analyses, the oil was dried over anhydrous sodium sulfate and kept in a sealed vial under N_2_ at 4 °C. An EO content of 1.9% *v*/*w* was found in the Biotype BIO 59.

### 2.4. Chemicals and Reagents

The generic caspase inhibitor z-VAD-FMK, purchased from Cell Signaling Technology (Beverly, MA, USA), was diluted in DMSO to a 10 mM concentration. For the experiments, this solution was diluted in a culture medium and used at a concentration of 50 µM.

Unless otherwise specified, the compounds used were purchased from Sigma-Aldrich (Milan, Italy).

For each experiment, a control test referring to cells treated with only the vehicle is reported.

Before use, oregano essential oil (OEO) was diluted in 1:1 DMSO and then diluted in a culture medium. In each experimental condition, DMSO never exceeded 0.04%, and it did not change cell viability. 

### 2.5. Cell Cultures

Human breast cancer epithelial cell line estrogen (ERα)-positive receptor MCF-7 and triple-negative human breast cancer cell line MDA-MB-231 (ER-, PR-, HER-2-negative) were purchased from “Istituto Scientifico Tumori” (Genoa, Italy). They were grown as monolayers in DMEM medium supplemented with 10% (*v*/*v*) heat-inactivated fetal bovine serum (FBS), antibiotic–antimycotic (100 U/mL penicillin, 100 μg/mL streptomycin, and 250 ng/mL amphotericin B), 2 mM glutamine, and 1% non-essential amino acids in a humidified atmosphere containing 5% CO_2_ at 37 °C. Cells were detached from the substrate using 10X trypsin-EDTA solution (5 mg/mL trypsin and 2 mg/mL EDTA) diluted in PBS (phosphate-buffered saline: 137 mM NaCl, 2.68 mM KCl, 10 mM Na_2_HPO_4_, KH_2_PO_4_ 1.76 mM, pH 7.4) at 0.25X concentration. Cells were seeded in 6 (2 × 10^5^ cells/2 mL), 24 (6 × 10^4^ cells/1 mL), and 96 well plates (8 × 10^3^ cells/200 µL). After plating, the cells were placed for 24 h in an incubator at 37 °C and subsequently treated with different compounds at various times, as indicated in the results.

All the reagents used for the cell cultures were purchased from Biosigma (Cona, VE, Italy). 

### 2.6. Assessment of Cell Viability and Morphology

Cell viability after the different treatments was evaluated using the MTT assay.

MTT (3-(4,5dimethylthiazol-2-yl)-2,5-diphenyl-2H-tetrazolium bromide) is a yellow tetrazolium salt that is reduced to formazan, blue–violet, by mitochondrial dehydrogenases.

Cells (8 × 10^3^ cells/200 µL culture medium) were seeded in 96-well plates and treated with increasing percentages of OEO in the presence or absence of z-VAD-FMK, a general caspase inhibitor, or 1,2-bis (o-aminophenoxy) ethane-N, N, N′, N′-tetraacetic acid (BAPTA), a calcium chelator. At the end of the treatment period, 20 µL of MTT (5.5 mg/mL of PBS) were added to each well. After 2 h of incubation at 37 °C, the medium was removed, and 100 µL of lysis buffer (20% sodium dodecyl sulfate, 40% N,N-dimethylformamide, pH 4.7) was added. Subsequently, we proceeded with the spectrophotometric reading using an ELISA reader (Opsys MR; Dynex Technologies, Chantilly, VA, USA), set with the following parameters: absorbance at 540 nm (test) and 630 nm (reference wavelength). The data obtained were reported as a percentage of residual viability with respect to the control, which is considered equal to 100% viability.

The morphological changes induced by OEO treatment were evaluated using an OPTICA inverted microscope (OPTIKA S.r.l., Ponteranica (BG), Italy).

IC_50_ was calculated employing the “Quest Graph™ IC_50_ Calculator.” AAT Bioquest, Inc., Pleasanton, CA, USA, https://www.aatbio.com/tools/ic50-calculator, accessed on 10 November 2023. 

Furthermore, Hoechst 33342 was used to highlight induced changes in chromatin. To evaluate the effects of the different dilutions of OEO, the cells were seeded in 24-well plates (6 × 10^4^ cells/1 mL of culture medium); after 24 h, they were incubated with Hoechst (2 µg/mL) at 37 °C for 10 min. Subsequently, the cells were treated with OEO and observed after 24 and 48 h using an OPTIKA IM3FL4 fluorescence microscope equipped with a digital imaging camera system (OPTIKA) using a Dapi (excitation wavelength of 355 nm and emission wavelength of 435 nm) filter. Images were taken at a magnification of 200 or 400X and acquired by a computer imaging system (OPTIKA PROVIEW, version x64, 4.11.20805.20220506).

### 2.7. Evaluation of the Mitochondrial Membrane Potential

For the evaluation of the mitochondrial membrane potential difference (ΔΨm), the lipophilic cationic fluorochrome JC-1 (5,5′,6,6′-Tetrachloro-1,1′,3,3′-tetraethylbenzimidazolylcarbocyanine iodide, Invitrogen by Thermo Fisher Scientific, Inchinnan Business Park, Paisley, PA4 9RF, UK) was used.

Cells were seeded in 24-well plates (6 × 10^4^ cells/1 mL of medium). Subsequently, they were incubated for 24 h with OEO. At the end of the treatment, the cells were incubated at 37 °C for 10 min with 5 µg/mL JC-1. At the end of the incubation, the culture medium containing the fluorochrome was removed, and the cells were observed under a fluorescence microscope (OPTIKA).

Red fluorescence (J-aggregates), detected by a “rhodamine filter” with a 596 nm excitation peak and a 620 nm emission peak, is more present in viable, healthy cells with mitochondria that are polarized and have high membrane potential. Green fluorescence (monomeric form of JC-1), detected by a “FITC filter” with a 485 nm excitation peak and a 530 nm emission peak, is prevalent in apoptotic cells with a low membrane potential.

The images shown are the result of the merge of the two fluorescences (OPTIKA PROVIEW, version x64, 4.11.20805.20220506).

The loss of Δψm was also quantified using 3,3-dihexyloxacarbocyanine (DiOC6 Molecular Probes, Eugene, OR), a cell-permeant green fluorescent, lipophilic dye that accumulates in the mitochondrial matrix when Δψm is intact. After treatment with OEO, cells (2 × 10^5^/well) were incubated with 40 nM DiOC6 for 20 min at 37 °C and analyzed by flow cytometry on a FACSCantoTM II (BD Biosciences Company, Franklin Lakes, NJ, USA) with an excitation peak at 488 nm and an emission peak at 525 nm. Data analysis was performed by FlowJo v10 software (BD Biosciences).

### 2.8. Intracellular Calcium Assessment

The increase in intracellular calcium levels was analyzed by the fluorescent dye Ca^2+^ FLUO 2-AM. After incubation with the compounds, the cells (2 × 10^5^/well) were washed in PBS and incubated with 1 ug/mL of FLUO 2-AM for 20 min at RT and in the dark. At the end of the incubation, the cells were harvested, and the fluorescence was analyzed by flow cytometry on a FACSCantoTM II (BD Biosciences) and analyzed using FlowJo v10 software (BD Biosciences).

### 2.9. Western Blotting Analysis

The cells were seeded in 6-well plates (2 × 10^5^ cells/2 mL), and following the treatment with the OEO, they were collected and centrifuged at 180 g for 8 min. Subsequently, they were lysed, keeping them at 4 °C, with suitable volumes of RIPA buffer (PBS plus 1% Nonidet P40, 0.5% sodium deoxycholate, 0.1% SDS, and protease inhibitors, pH 7.4). After sonication (Soniprep 150, Cellai Srl, Milan, Italy), the protein content was evaluated by the Bradford assay. The samples (30 µg of proteins) were inactivated at 95 °C for 5 min and subjected to electrophoretic separation in SDS-PAGE, together with a standard of known molecular weight. Tris-glycine buffer (glycine 94 g, SDS 5 g, pH 8.3) was used for the electrophoretic flow. Then, the proteins were blotted on a nitrocellulose membrane in the presence of transfer buffer (48 mM tris base, 39 mM glycine, 20% methanol, and 0.037% SDS) and then incubated with a blocking solution (milk solubilized in TBST, 20 mM Tris-HCl, 150 mM NaCl, pH 7.5, 0.05% Tween 20) for 1 h. The membrane was then incubated at 4 °C overnight (ON) with the primary antibody directed against the protein of interest (1 ug/1 mL TBST): Bax and Bcl-2 (Santa Cruz Biotechnology, Santa Cruz, CA, USA), caspases 3 and 9 (Cell Signaling, Beverly, MA, USA), and β-actin (Sigma-Aldrich, St. Louis, MO, USA).

Subsequently, the filter was washed three times in TBST and incubated for one hour with the secondary antibody (1 µg/5 mL TBST) conjugated with horseradish peroxidase. The signal obtained by development in enhanced chemiluminescence (ECL) was detected by CHEMIDOC (Bio-Rad Laboratories, Hercules, CA, USA). The bands obtained were quantified by densitometric analysis with the software Quantity One, version 4.6.6 (BioRad, Hercules, CA, USA), normalizing the values with respect to the constitutive protein actin.

After development, the membrane was subjected to stripping (15 g glycine pH 2.2, 1 g SDS, 10 mL Tween) to allow further evaluation.

### 2.10. Intracellular ROS

Intracellular ROS were detected using the redox-sensitive fluorochrome 2′,7′-dichlorodihydrofluorescein diacetate (H_2_-DCFDA, Molecular Probe, Life Technologies, Eugene, OR, USA). H_2_-DCFDA diffuses into cells and is deacetylated by cellular esterases to form 2′,7′-dichlorodihydrofluorescein (H_2_DCF). In the presence of ROS, H_2_DCF is oxidized to 2′,7′-dichlorofluorescein (DCF), which emits green fluorescence when appropriately excited. Cells (6 × 10^4^/well) were seeded in 24-well plates and, after 24 h, were incubated with 0.05% OEO at different times. In the end, the cells were washed with PBS and incubated with 1 μM H_2_-DCFDA in Hank’s balanced salt solution (HBSS) for 30 min at 37 °C in darkness. Then, H_2_-DCFDA was removed and replaced with HBSS. Fluorescence images were acquired with a fluorescence microscope (OPTIKA) equipped with a digital imaging camera system (OPTIKA) using a FITC filter (excitation wavelength of 485 nm and emission wavelength of 530 nm). Images were taken at a magnification of 200 and acquired by a computer imaging system (OPTIKA PROVIEW, version x64, 4.11.20805.20220506).

### 2.11. DPPH Assay

The 1,1-diphenyl-2-picrylhydrazyl (DPPH) radical scavenging activity method was used to evaluate the antioxidant potential of OEO [25]. The DPPH has a maximum absorbance of 517 nm; the decrease in the peak is indicative of the reduction of the DPPH due to the presence of antioxidant compounds.

A total of 100 µM of DPPH solubilized in ethanol was added to different percentages (0.025% and 0.05%) of OEO in a final volume of 1 mL. After incubation for 30 min at RT, the absorbance of the sample at 517 nm (AOEO) was measured by spectrophotometry. The absorbance of the sample was referred to as containing only DPPH (Blank sample, AB). The data were expressed as radical scavenging activity (RSA) and calculated as follows [26]:RSA (%) = ((AB − AOEO)/AB) × 100

### 2.12. Isolation of Volatile Components

EOs were extracted in accordance with Basile et al. [27]. Air-dried samples, grounded with a Waring blender, were hydrodistillated for 3 h in accordance with the standard procedure described in European Pharmacopoeia [24]. Anhydrous sodium sulfate was used to dry the oil, which was then stored in a sealed vial under N_2_ at −20 °C, ready for GC–MS analyses.

### 2.13. X.2. GC–MS Analysis

The analysis of EO was performed according to the procedure reported by Bancheva et al. (2022) [28]. GC–MS analysis of EO was performed using an Agilent 7000C equipped with an apolar capillary column (DB-5MS) in fused silica (30 m × 0.25 mm i.d.; 0.25 μm film thickness) (Santa Clara, CA, USA) coupled to an MSD 5973 triple quadrupole detector (Mass Selective Agilent). The oven program was as follows: temperature increase at 40 °C for 5 min, at a rate of 2 °C/min up to 260 °C, then isothermal for 20 min. Helium was used as carrier gas (1 mL min^−1^). The injector and detector temperatures were set at 250 °C and 290 °C, respectively. A total of 1 μL of oil solution (3% EO/Hexane *v*/*v*) was injected with split mode 1.0; MS range 40–600. The percentages in Table 3 are calculated with the TIC from MS. The analysis was performed with the following settings: ionization voltage, 70 eV; electron multiplier energy, 2000 V; transfer line temperature, 295 °C; solvent delay, 4 min. Linear retention indices (LRI) were determined by employing retention times of n-alkanes (C8–C40). Identification of peaks was conducted by comparison with mass spectra and comparison of the relative retention indices with the WILEY275, NIST 17, ADAMS, and FFNSC2 libraries.

### 2.14. Statistical Analysis

The results were represented as mean ± standard deviations (S.D.). Statistical analysis was performed using the GraphPadPrismTM 4.0 software (GraphPad Prism^TM^ Software Inc., San Diego, CA, USA). Data were analyzed using the Student’s *t*-test and one-way ANOVA, followed by the Bonferroni multiple comparisons test. Differences were considered significant when *p* < 0.05, and one-way ANOVA was the statistical method used in data evaluation. Differences were considered significant at *p* < 0.05.

## 3. Results

### 3.1. The Effects of Oregano Essential Oil (OEO) on Cell Viability 

Initially, we evaluated the effect of oregano essential oil (OEO) on cell viability using an MTT colorimetric assay. MDA-MB-231 and MCF-7 cells were treated for 24 h and 48 h with increasing percentages of OEO. Histograms reported in Figure 1A showed that OEO reduces cell viability in a dose- and time-dependent manner. After 24 h of treatment with 0.05% OEO, the residual viability was about 50% in both MDA-MB-231 and MCF-7 cells. After 48 h of treatment, the effect of OEO in both cell lines was even more evident, reaching residual values close to 10% with 0.2% OEO (Figure 1A). Table 1 shows the reported IC_50_ values of OEO in both cell lines at 24 h and 48 h.

Based on the results obtained, all the subsequent analyses were performed using 0.025% or 0.05% OEO, doses that have proven effective but non-toxic.

Table 1 IC_50_ data were obtained after 24 h and 48 h of treatment of MDA-MB-231 and MCF-7 cell lines. IC_50_ was calculated using the “Quest Graph™ IC_50_ Calculator.” AAT Bioquest, Inc., https://www.aatbio.com/tools/ic50-calculator, accessed on 11 November 2023, as reported in Section 2. KIs based on literature (https://webbook.nist.gov/, accessed on 11 November 2023); b Experimental KIs on a DB-5MS apolar column; components were listed in order of elution on a DB-5MS apolar column.

Optical microscope observations confirm that 0.05% of OEO reduced the number of MDA-MB-231 and MCF-7 (Figure 1B), as well as induced morphological changes that suggest the induction of cell death. 

Cell death activation by OEO treatment was demonstrated by staining the cells with the fluorochrome Hoechst 3342. 

As shown in Figure 1C, the nuclei of untreated breast cancer cells appear spherical, with DNA distributed uniformly. After 24 h of treatment with 0.05% OEO, condensed, crescent-shaped chromatin in the nuclei was observed in both MDA-MB-231 and MCF-7 cells. These effects are more evident after 48 h of treatment.

### 3.2. The Effects of Oregano Essential Oil (OEO) on Caspase Activation

To clarify the death mechanism activated by OEO in breast cancer (BC) cells, we evaluated the possible involvement of caspases. Our data demonstrated that z-VAD-FMK, a general caspase inhibitor, partially reduced the cytotoxic effect induced by OEO in BC cells, thus suggesting the involvement of caspases in the death mechanism. In fact, as shown in Figure 2A, the addition of z-VAD-FMK reduced the OEO cytotoxic effect by 45% in MDA-MB-231 and MCF-7 cells. 

Western blotting experiments were thus performed to study the main caspases involved in the mechanism of death. The results showed that OEO treatment did not induce pro-caspase-8 activation. In fact, no significant changes in the full-length form or the appearance of active fragments of caspase-8 were observed in the cell lines examined (Figure 2B). Instead, as shown in Figure 2B, treatment with OEO determined the activation of pro-caspase-9 and caspase-3. In fact, after 24 h of treatment with OEO, a decrease in the levels of both caspases and the appearance of the active fragments was found in both MDA-MB-231 and MCF-7 cells. These effects were already present at 0.025% OEO, becoming more evident at 0.05%. Furthermore, the activation of caspase-3 induced by OEO was also confirmed by the cleavage of PARP (Figure 2B), one of its targets [29]. 

### 3.3. OEO Effects on Mitochondria and Bcl-2 Family Members

To evaluate the possible involvement of the mitochondria in the mechanism of cell death induced by OEO, we evaluated the mitochondrial membrane potential (Ψm) using the lipophilic cationic fluorescent dye JC-1 (5,5′,6,6′-tetrachloro-1,1′,3,3′-tetraethylbenzimidazolylcarbocyanine iodide [30].

As shown in Figure 3A, in untreated MDA-MB-231 and MCF-7 cells, JC1 is predominantly present in the form of red J-aggregates, indicating an intact ΔΨm. Instead, treatment for 24 h with OEO with 0.05% of both BC cell lines determined an increase in the green fluorescence compared to the red one, indicating a loss of the ΔΨm. 

The loss of ΔΨm induced by 0.05% OEO was further quantified by flow cytometric analysis using the fluorochrome DiOC6. As can be observed in Figure 3B, a high fluorescence peak appears in the untreated MDA-MB-231 and MCF-7 cells. Treatment for 24 h with 0.05% OEO induces a shift of the peak towards values of lower fluorescence in both cell lines, confirming the loss of ΔΨm.

One of the causes of the loss of ΔΨm can be attributed to the formation of the mitochondrial permeability transition pore (PTP) [31]. The proapoptotic factor Bax and the voltage-dependent anion channel (VDAC) are involved in the formation of the PTP, while the antiapoptotic factors of the Bcl-2 family inhibit its formation [32,33,34]. Western blotting analysis showed that 24 h treatment with 0.05% OEO increased in both MDA-MB-231 and MCF-7 cells the levels of Bax and VDAC while decreasing the levels of the antiapoptotic factors Bcl-2 and Bcl-xL (Figure 3C). Interestingly, especially in MCF-7, treatment with OEO determined the appearance of a high molecular weight band (about 55 kDa), probably corresponding to an oligomeric form [35,36].

### 3.4. OEO Effect on ROS Production and Intracellular Ca^2+^ Level

Tumor cells show a sustained increase in ROS production, which appears to promote tumor progression. On the other hand, the induction of ROS by chemotherapeutics seems to be a useful mechanism to induce cell death [37].

The analysis of ROS production using the fluorochrome H_2_-DCFDA highlighted that OEO treatment did not induce ROS production in the first hours of treatment. Instead, an increase in ROS was evident at 24 h of treatment (Figure 4A), when the cytotoxic effect was not yet evident.

It has been shown that different phytocompounds present in EOs possess antioxidant properties [38]. To ascertain the antioxidant activity of OEO, we performed a DPPH radical scavenging assay, as reported in Section 2. Table 2 shows the effect of different percentages (0.025% and 0.05%) of OEO. Our data demonstrated that OEO possesses elevated and dose-dependent radical scavenging activity. In fact, the percentage of the antioxidant activity exerted by 0.025% OEO was only 26.82%, and this effect increased to 43.9% with 0.05% OEO. Superoxide dismutase (SOD2) and catalase are two important ROS scavenger enzymes [39]. We demonstrated that treatment for 24 h with 0.025% of OEO upregulated the levels of both enzymes in MDA-MB-231 and MCF-7. Instead, treatment with 0.05% OEO induced a significant increase only in catalase in MDA-MB-231 (Figure 4B).

Ca^2+^ is a second messenger that often intervenes in the activation of cell death [40]. Many natural components present in EO can induce an increase in cytosolic calcium. For this reason, we evaluated whether OEO induces changes in cytosolic calcium concentration. Flow cytometric data, obtained using fluorescent dye Fluo-2AM, showed that only after 24 h of treatment with 0.05% of OEO, there was an increase of about 17% of calcium in the cytosol, both in MDA-MB-231 and in MCF-7 cells (Figure 5A). To understand if this variation could affect cell viability, we performed cell viability experiments using BAPTA, a calcium chelator. The results showed that the combined treatment of 0.05% OEO and 5 µM BAPTA did not prevent the OEO-induced cytotoxic effect (Figure 5B).

### 3.5. Evaluation of OEO Composition and Effect on Cell Viability of the Main Constituents

The chemical components of OEO change according to the geographical area of growth, the genetic variations of the species, the cultivation practices, and the environmental conditions [9,41]. To evaluate the chemical components of the OEO used for our study, GC–MS analysis was performed. Twenty-six compounds were identified and listed in Table 3 according to their retention indices on a DB-5Ms apolar column. 

Among the compounds identified, *p*-cymene, *γ*-terpinene, thymoquinone, and p-acetanisole have attracted our attention due to the anticancer properties described in the literature [42,43,44,45,46]. Table 4 shows the reported IC_50_ values of the single compounds. Therefore, cell viability studies in MDA-MB-231 and MCF-7 cells were performed using combinations of these compounds in the same percentages found in OEO. The results seem to indicate that these compounds used alone have a modest effect on cell viability, while when used together, they significantly reduce cell viability in both MDA-MB-231 and MCF-7 cell lines in a time and dose-dependent manner, almost analogous to OEO (Figure 6). Interestingly, the cytotoxic effects exerted by the combination on BC cells were like those exerted by OEO. 

Furthermore, we evaluated the possible loss in ΔΨm using the same percentages of *p*-cymene, *γ*-terpinene, thymoquinone, and p-acetanisole used alone and in association with the concentration present in the 0.05% OEO. We have highlighted that in both MDA-MB-231 and MCF-7 cells, only in association is clearly visible the increase in the green fluorescence compared to the red index of loss of the ΔΨm (Figure 7).

## 4. Discussion

The most problematic aspects of anticancer therapies are toxic side effects [47] and the onset of multidrug resistance [48]. Therefore, research in the oncological field is aimed at finding new chemotherapy drugs that are increasingly efficient and targeted, causing less toxicity in normal cells. In recent years, interest in anticancer agents and adjuvant therapy has turned to compounds of natural origin, including essential oils (EOs) [17,49,50]. EOs are mixtures of various organic, volatile, and oily substances, consisting of terpenes, monoterpenes, sesquiterpenes, phenolic compounds, volatile compounds, and low molecular weight [5,17,51,52]. They have long been used in folk medicine; however, their biological properties have been demonstrated in recent years. In fact, experimental evidence has highlighted that EOs are able to exert various activities, including antibacterial, antiviral, antioxidant, anti-inflammatory, and antitumoral properties [13,15,53,54,55,56,57].

In our study, we explored the antitumoral effects of EO from Oregano (OEO) grown in Sicily on breast cancer cells. Oregano, scientific name *Origanum vulgare* L., is a spice native to the arid, rocky soils of the Mediterranean, also present in the Euro/Iranian-Siberian regions [1,9,41]. This spice is used as a fresh or dry extract, and its EO is used both in the food sector and for its therapeutic applications due to its antioxidant and antibacterial properties [18,20,21,23].

Here, we provide evidence for the first time that OEO has strong effects on reducing the viability of two breast cancer (BC) cells: MDA-MB-231, a triple-negative breast cancer (TNBC) cell line, and MCF-7, a cell line responsive to estrogen. Notably, 16HBE14o, a non-tumor human bronchial epithelial cell line, appeared to be more resistant to OEO treatment (see Appendix A), thus suggesting a reduced cytotoxic effect of OEO in normal cells than in tumor ones.

To understand the molecular mechanism underlying the cytotoxic effect of OEO, we then focused our attention on the possible activation of caspases. Our data demonstrated that OEO did not activate caspase-8, a caspase involved in the extrinsic apoptotic pathway [58], while induced fragmentation of pro-caspase-9 and 3 with the production of the active form of these enzymes as well as the breakdown of PARP, a specific substrate of caspase-3 [29]. These data support the conclusion that OEO exerts cytotoxic effects in BC cells by activating intrinsic apoptotic cell death. Notably, caspase activation plays a crucial role in OEO-induced apoptosis, as suggested by the finding that the addition of z-VAD, a cell-permeable, broad-spectrum inhibitor of caspase activity, partially protected BC cells from the loss of viability induced by OEO.

Interestingly, our results also indicated that mitochondrial dysfunction plays an important role in the mechanism of death activated by OEO. Notably, we observed that OEO caused depolarization of the mitochondrial transmembrane potential both in MDA-MB-231 and MCF-7 cells. These findings were in accordance with the study of Alves et al. [59], demonstrating that OEO exerts an antiparasitic effect by inducing mitochondrial depolarization.

The proteins of the Bcl-2 family are essential factors in the control of mitochondrial integrity [33,60]. In both MDA-MB-231 and MCF-7 cells, treatment with OEO provoked a decrease in the levels of the antiapoptotic factors Bcl-2 and Bcl-xL while increasing the levels of Bax. It is possible that changes in the expression of the Bcl-2 family can contribute to the loss of mitochondrial membrane potential induced by OEO. 

Our results also demonstrated that OEO treatment induced in BC cells an increase in calcium levels. The observation that the addition of BAPTA, a calcium chelator, did not reduce the OEO cytotoxic effects in BC cells suggests that the calcium increase is a consequence and not the cause of OEO-induced cell death. The origin of the calcium increase is unknown at this time. Changes in intracellular calcium content can be attributed to voltage-gated calcium channels opening in response to the falling transmembrane potential and/or the increasing intracellular sodium concentration through calcium membrane transporters [61]. The observation that OEO treatment in breast cancer cells upregulated Bax while lowering VDAC levels supports the conclusion that cytosolic calcium increases could originate from mitochondria.

Many chemotherapeutics act through oxidative stress [62]. Oxidative stress is induced not only in cancer cells but also in normal cells [63]. On the one hand, this effect induces cell death, but on the other, it can lead to the activation of chemoresistance mechanisms [64,65]. We evaluated whether the cytotoxic effect promoted by OEO was linked to oxidative stress and demonstrated that it increased ROS levels. Furthermore, Western blotting analyses show that OEO upregulated the antioxidant enzymes SOD2 and catalase. We could hypothesize that OEO-induced death does depend on ROS production, a late event that can result from mitochondrial damage. Indeed, OEO seems to play an antioxidant role. This is in line with the bibliographic data, which show that OEO, thanks to its chemical components, plays an antioxidant role [4,66,67]. Although the antioxidant effect of OEO can slow down death, it can be useful to reduce the toxic damage of chemotherapy. 

The chemical composition of EOs changes in relation to the pedoclimatic conditions of cultivation. Interestingly, the characterization of the polyphenolic profile of OEO using HPLC/MS provided evidence that OEO contains different compounds with antioxidant and antitumor potential [4,14,56]. Among the most abundant compounds, there is p-acetanisole, which has antioxidant potential [68]. Of particular biological interest are also *p*-cymene, *γ*-terpinene, and thymoquinone, with potentially different biological roles such as antioxidant, anti-inflammatory, and antitumor [42,43,44,45,46].

To understand the contribution of the different phytochemicals found in OEO to the cytotoxic effect exerted on BC, we tested the effects of the main components present in OEO alone or in combination to explore possible synergistic interactions. We demonstrated that *p*-cymene, *γ*-terpinene, thymoquinone, and p-acetanisole did not exert a significant cytotoxic effect when tested alone. At the same time, when added together, they can reduce cell viability with loss of ΔΨm, effects like OEO. This finding suggests that the different compounds contained in OEO act in synergy.

## 5. Conclusions

Taken together, our findings demonstrate the anticancer activities of Sicilian *Origanum vulgare* L. essential oil, which represents a resource of bioactive compounds such as *p*-cymene, *γ*-terpinene, thymoquinone, and *p*-acetanisole. However, further experiments need to be conducted to detail the molecular pathway activated by oil compounds and the effect of OEO on other cancer and non-tumor cell lines (Figure 8). It could also be extremely interesting to evaluate the association of OEO or its components with classical chemotherapy to assess their potential beneficial effects and the possible reduction in their toxic side effects and multi-resistance.

For future clinical studies on humans, in addition to evaluating the safety and toxicity of OEO, it will be necessary to formulate and standardize clinical protocols for its administration.

## Figures and Tables

**Figure 1 cells-12-02733-f001:**
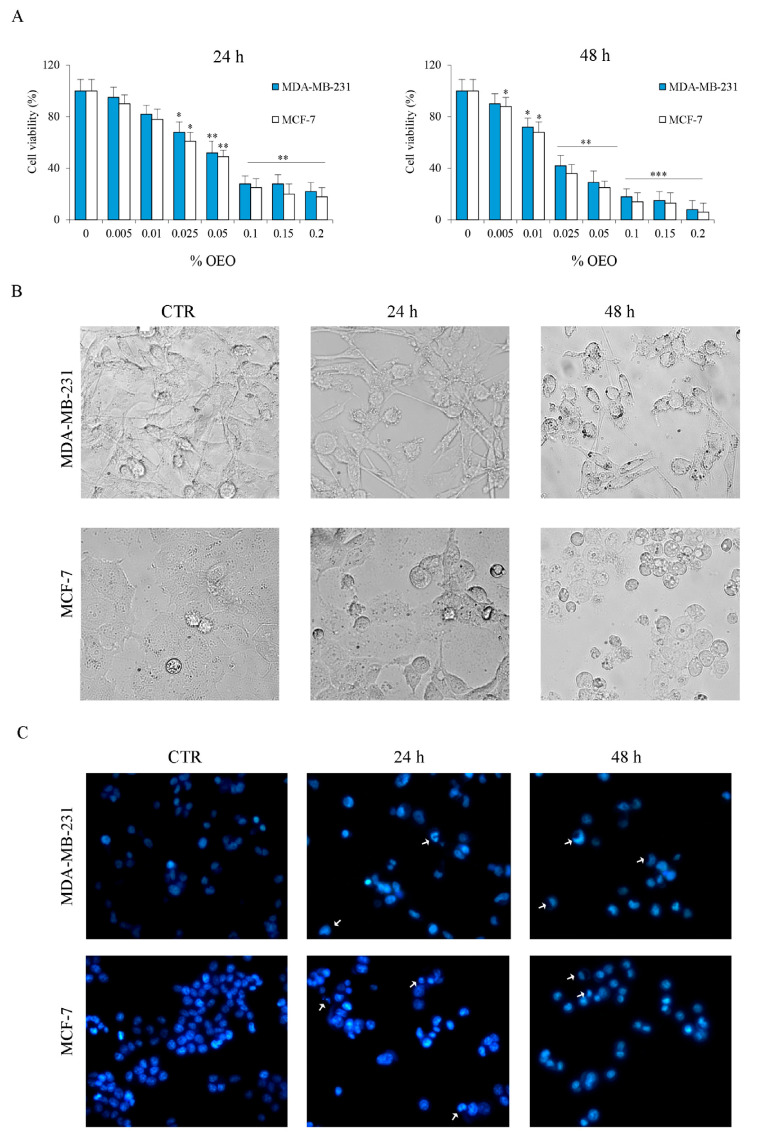
The cytotoxic effect exerted by OEO on MDA-MB-231 and MCF-7 cells. (**A**) Effects of OEO on cell viability. Cells were treated with different percentages of OEO for 24 h and 48 h. Cell viability was assessed by MTT, as described in Section 2. (**B**) Morphologic changes in MDA-MB-231 and MCF-7 cells were observed under (**B**) light microscopy and (**C**) fluorescence microscopy after Hoechst33342 staining. Images were taken at 200X magnification. Arrows indicate OEO-induced changes in chromatin. Results are representative of three independent experiments. In (**A**), values are the means of three independent experiments ± S.D. * *p* < 0.05, ** *p* < 0.01, *** *p* < 0.001 vs. untreated control.

**Figure 2 cells-12-02733-f002:**
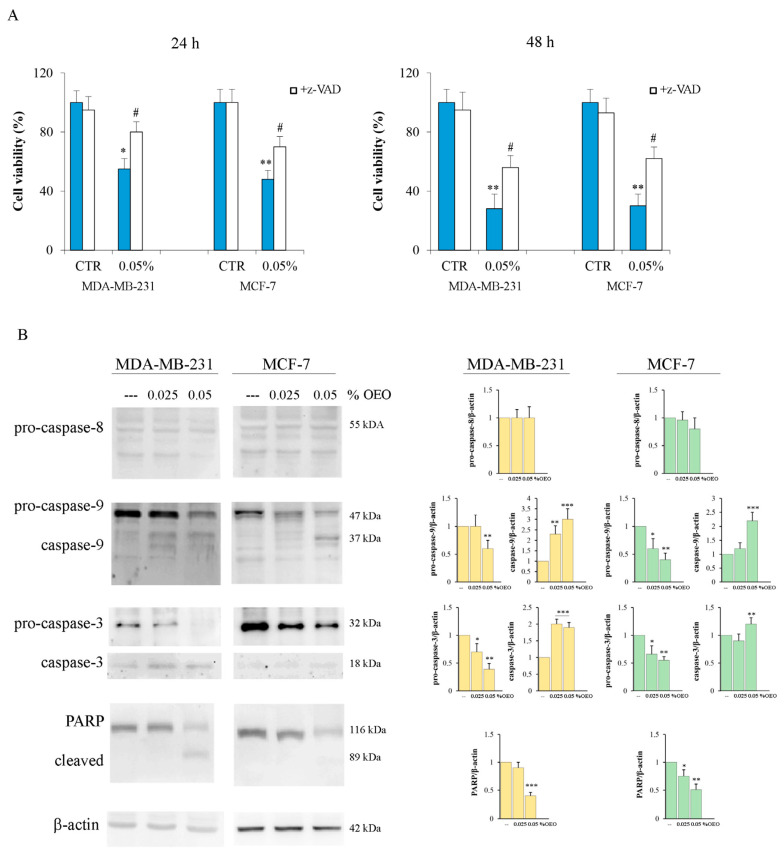
OEO-induced caspase activation. (**A**) The caspase inhibitor z-VAD-FMK reduced the OEO cytotoxic effect in BC cells. MDA-MB-231 and MCF-7 cells were treated with 0.05% OEO without or with 50 µM z-VAD-FMK for 24 h and 48 h. Cell viability was assessed by MTT, as described in Section 2. (**B**) The effect of OEO on pro-caspase activation. MDA-MB-231 and MCF-7 cells were treated for 24 h with 0.025 or 0.05% OEO and then evaluated for the level of pro-caspases 8, -9, and -3 and their cleavage forms by Western blotting analysis. The data show the densitometric analyses obtained using the Quantity One software. The protein content was normalized with respect to β-actin; the reported values, the mean of three independent experiments, are reported with respect to the untreated control, which was assigned a value of 1. In (**A**,**B**), values are the means of three independent experiments ± S.D. * *p* < 0.05, ** *p* < 0.01; *** *p* < 0.001 vs. untreated control; # *p* < 0.05 vs. treated cells.

**Figure 3 cells-12-02733-f003:**
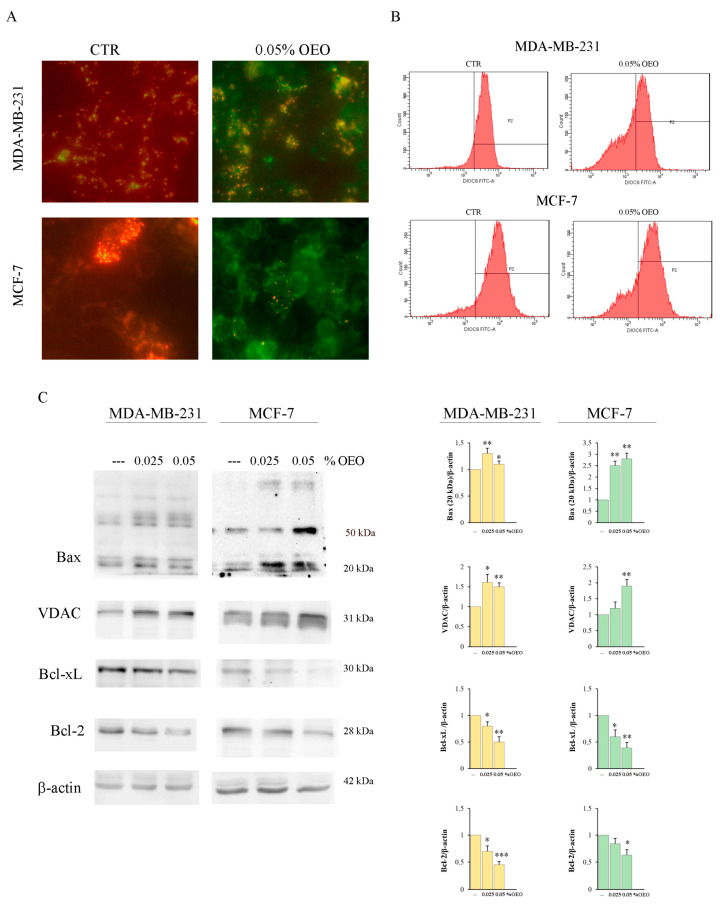
The effects of OEO on mitochondrial membrane potential (ΔΨm) and levels of Bcl-2 family members and VDAC. (**A**,**B**) OEO-induced mitochondrial depolarization in MDA-MB-231 and MCF-7 cells. Cells were treated for 24 h with 0.05% OEO. (**A**) In the end, cells were incubated with JC-1 fluorochrome, and fluorescent cells were visualized with an OPTIKA microscope at 400X magnification, as indicated in Section 2. (**B**) Dissipation of ΔΨm was also evaluated by flow cytometry using the lipophilic dye DiOC6. (**C**) The effect of OEO on the BCl-2 family and VDAC. MDA-MB-231 and MCF-7 cells were treated for 24 h with 0.025 or 0.05% OEO, and then the levels of Bax, VDAC, Bcl-xL, and Bcl-2 were evaluated by Western blotting analysis. The data show the densitometric analyses obtained using the Quantity One software. The protein content was normalized with respect to β-actin; the reported values, the mean of three independent experiments, are reported with respect to the untreated control, which was assigned a value of 1. The results are representative of three independent experiments. In (**C**), the reported values are the means of three independent experiments ± S.D. * *p* < 0.05, ** *p* < 0.01; *** *p* < 0.001 vs. untreated control.

**Figure 4 cells-12-02733-f004:**
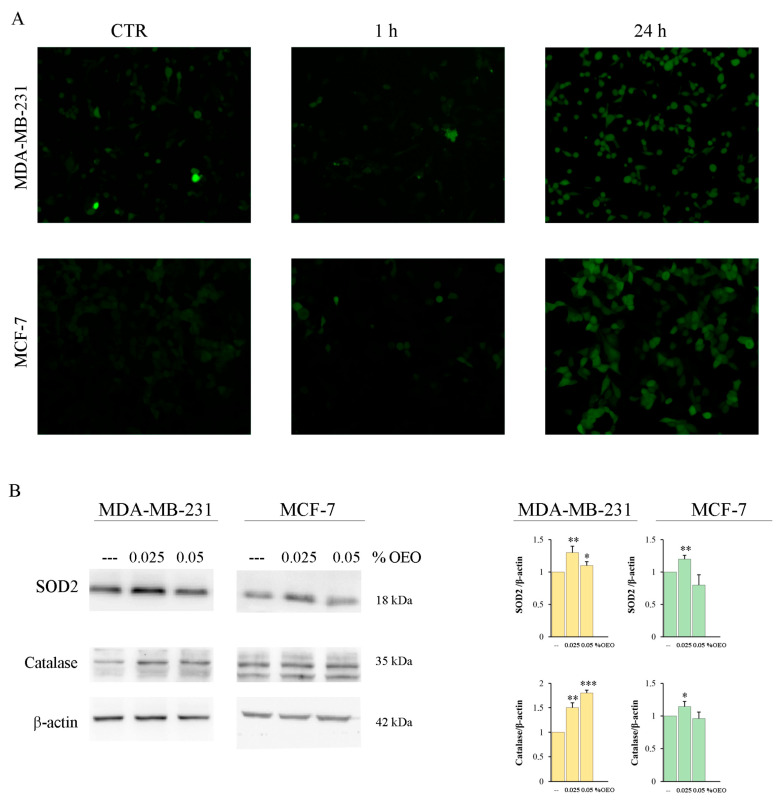
The effects of OEO on oxidative stress. (**A**) OEO induces ROS production. MDA-MB-231 and MCF-7 were treated for 1 h and 24 h with 0.05% OEO. In the end, cells were incubated with 1 µM H_2_-DCFDA, and fluorescent cells were visualized with an OPTIKA microscope at 200X magnification, as indicated in Section 2. (**B**) The effect of OEO on antioxidant enzymes. MDA-MB-231 and MCF-7 cells were treated for 24 h with 0.025 or 0.05% OEO, and then the levels of SOD2 and catalase were evaluated with Western blotting analysis. The data show the densitometric analyses obtained using the Quantity One software. The protein content was normalized with respect to β-actin; the reported values, the mean of three independent experiments, are reported with respect to the untreated control, which was assigned a value of 1. (**A**) The results are representative of three independent experiments. In (B), the reported values are the means of three independent experiments ± S.D. * *p* < 0.05, ** *p* < 0.01; *** *p* < 0.001 vs. untreated control.

**Figure 5 cells-12-02733-f005:**
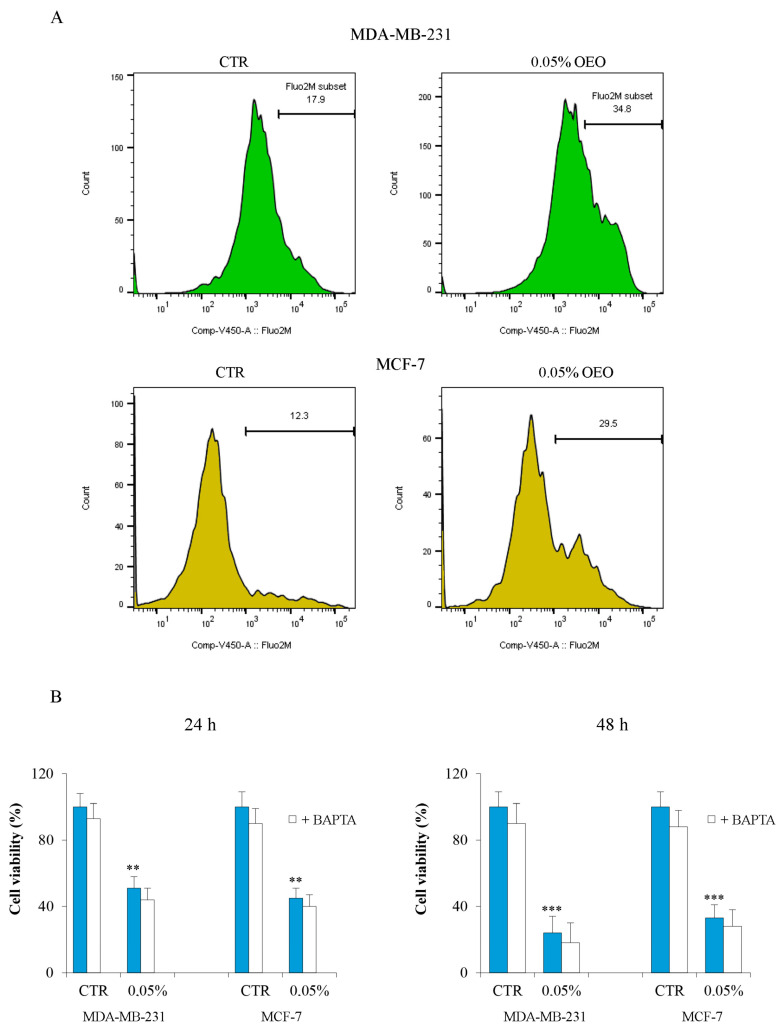
OEO treatment increased the intracellular calcium level. (**A**) MDA-MB-231 and MCF-7 cells were treated with 0.05% OEO for 24 h and 48 h. Variation in the content of intracellular calcium was evaluated by flow cytometry using Fluo 2-AM fluorochrome as reported in Section 2. (**B**) Effects of OEO/BAPTA treatment on cell viability. Cells were treated with 0.05% OEO without or with 5 µM BAPTA for 24 h and 48 h. Cell viability was assessed by MTT, as described in Section 2. In (**B**), values are the means of three independent experiments ± S.D. ** *p* < 0.01; *** *p* < 0.001 vs. untreated control.

**Figure 6 cells-12-02733-f006:**
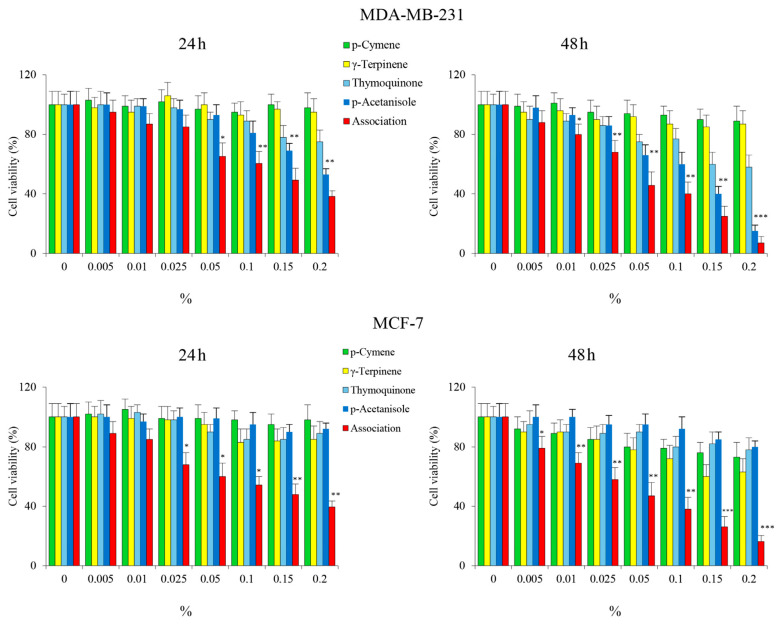
The cytotoxic effect exerted by some constituents of OEO on MDA-MB-231 and MCF-7 cells. Cells were treated with different percentages of *p*-cymene, *γ*-terpinene, thymoquinone, and p-acetanisole, employed alone or in association, for 24 h and 48 h. Cell viability was assessed by MTT, as described in Section 2. Results are representative of three independent experiments. Values are the means of three independent experiments ± S.D. * *p* < 0.05, ** *p* < 0.01, *** *p* < 0.001 vs. untreated control.

**Figure 7 cells-12-02733-f007:**
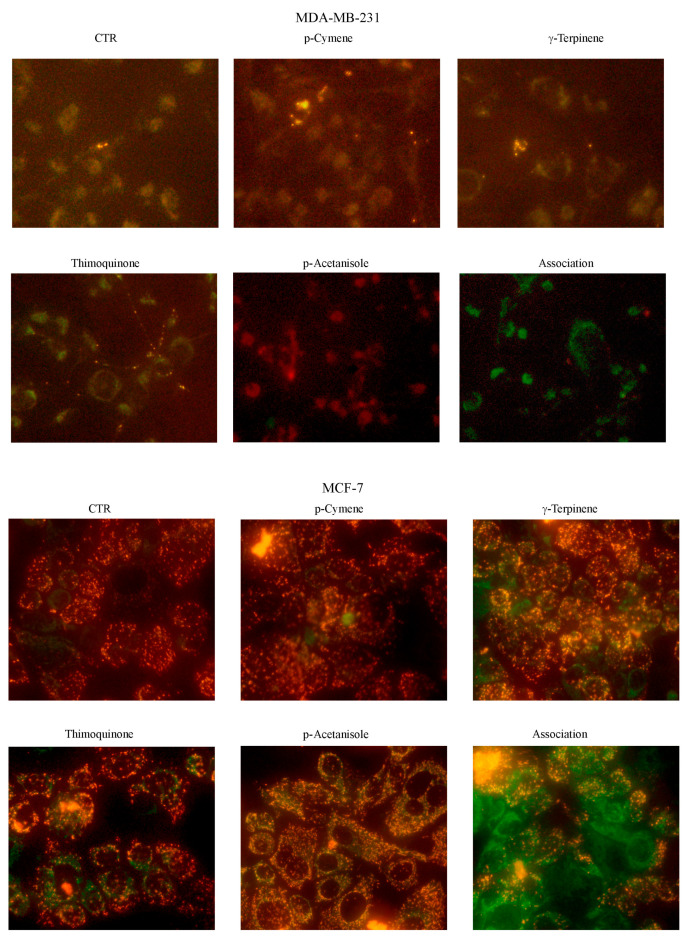
The effects of some constituents of OEO on mitochondrial membrane potential (ΔΨm). Cells were treated with *p*-cymene, *γ*-terpinene, thymoquinone, and p-acetanisole, employed alone or in association, for 24 h. In the end, cells were incubated with JC-1 fluorochrome, and fluorescent cells were visualized with an OPTIKA microscope at 400X magnification, as indicated in Section 2.

**Figure 8 cells-12-02733-f008:**
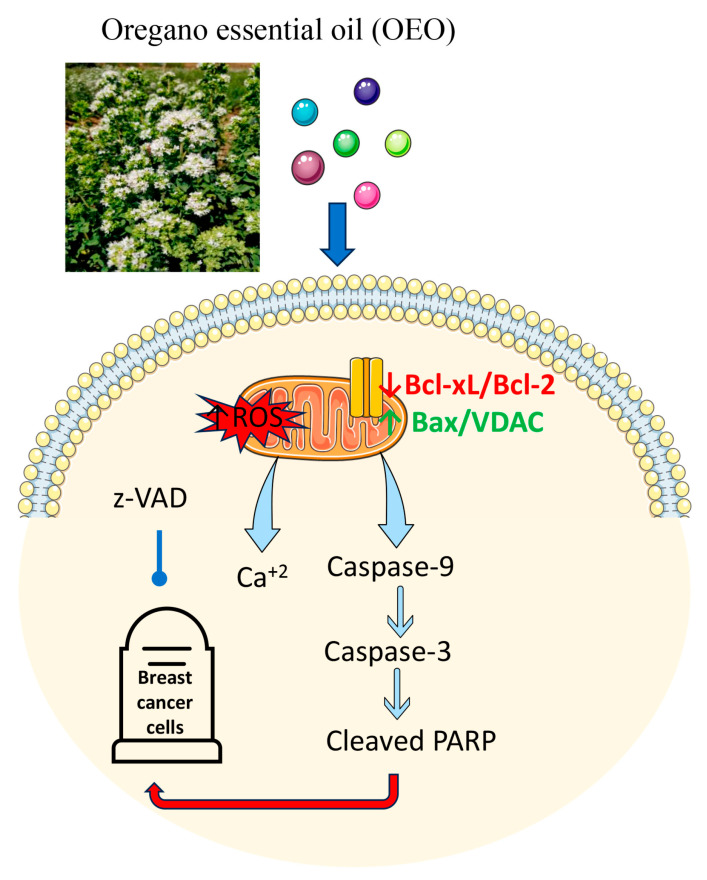
Schematic representation of the mechanism of action of OEO. Parts of the figure were drawn by using pictures from Servier Medical Art. Servier Medical Art by Servier is licensed under a Creative Commons Attribution 3.0 Unported License (https://creativecommons.org/licenses/by/3.0/ accessed on 19 September 2023).

**Table 1 cells-12-02733-t001:** OEO IC_50_.

24 h		48 h	
MDA-MB-231	MCF-7	MDA-MB-231	MCF-7
0.0542%	0.04%	0.022%	0.015%

**Table 2 cells-12-02733-t002:** Antioxidant activity of OEO.

Sample	Absorbance (517 nm)	% of Antioxidant Activity
DPPH	0.41	--
0.025% OEO	0.30 ± 0.05	26.82%
0.05% OEO	0.23 ± 0.07	43.9%

**Table 3 cells-12-02733-t003:** Composition of OEO.

	^a^ KI	^b^ AI	Compounds	RT	BIO59 (%)
1	927	922	*α*-Thujene	12.853	1.2
2	932	927	*α*-Pinene	13.153	0.9
3	946	940	Camphene	14.053	t
4	974	969	Sabinene	16.002	0.32
5	995	994	Myrcene	17.652	2.3
6	1004	1003	δ-3-Carene	18.302	0.89
7	1021	1018	α-Terpinene	19.351	4.16
8	1037	1033	*p*-Cymene	20.451	8.6
9	1047	1043	*α*-Phellandrene	21.151	0.93
10	1075	1072	γ-Terpinene	23.250	10.5
11	1079	1076	*cis*-sabinene hydrate	23.550	t
12	1090	1089	Terpinolene	24.500	0.64
13	1114	1113	Dehydrosabinaketone	26.200	1.15
14	1135	1133	allo-Ocimene	27.649	0.51
15	1187	1185	Terpinen-4-olo	31.448	1.3
16	1220	1219	Sabinene hydrate acetate	33.798	0.36
17	1243	1241	Carvacrol, methyl ether	35.297	4.2
18	1255	1253	Thymoquinone	36,147	5.12
19	1348	1346	p-Acetanisole	42.395	21
20	1354	1366	Thymol acetate	43.745	0.27
21	1371	1369	Carvacrol acetate	43.945	0.78
22	1416	1400	*β*-Longipinene	46.894	3.63
23	1453	1405	Italicene	49.093	0.29
24	1477	1407	Longifolene	50.593	1.05
25	1510	1509	Germacrene A	52.642	2.65
26	1523	1522	γ-Cadinene	53.392	5.13

^a^ KIs based on literature (https://webbook.nist.gov//, accessed on 19 November 2023); ^b^ Experimental KIs on a DB-5MS apolar column.

**Table 4 cells-12-02733-t004:** IC_50_.

Compounds	24 h		48 h	
	MDA-MB-231	MCF-7	MDA-MB-231	MCF-7
*p*-Cymene	0.4%	0.35%	0.28%	0.22%
γ-Terpinene	0.35%	0.31%	0.31%	0.28%
Thymoquinone	0.54%	0.57%	0.22%	0.31%
p-Acetanisole	0.04%	0.28%	0.025	0.12%

The IC_50_ values were obtained for *p*-Cymene, *γ*-Terpinene, thymoquinone, and p-acetanisole. IC_50_ was calculated using the “Quest Graph™ IC_50_ Calculator.” AAT Bioquest, Inc., https://www.aatbio.com/tools/ic50-calculator, accessed on 11 November 2023, as reported in Section 2.

## Data Availability

Data are contained within the article and Appendix A.

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
