# Peer review of "Cytotoxic Effect Induced by Sicilian Oregano Essential Oil in Human Breast Cancer Cells"

_cells, 2023, doi:10.3390/cells12232733_

Round 1

Reviewer 1 Report

Comments and Suggestions for Authors

Manuscript cells-2703156 presents the breast cancer treatment potential of Origanum vulgare essential oil. The authors have examined the cytotoxic activity of oregano essential oil on MCF-7 and MDA-MB-231 cells as well as possible mechanisms of cytotoxic activity.

Some major points the authors should address:

1.       Based on the data illustrated in Figure 1A, what are the 24-h and 48-h IC50 values?

2.       What are the IC50 values for p-cymene, γ-terpinene, and thymoquinone (Figure 6)? It would be helpful to determine IC50 values based on these data for easy comparison with others in the literature.

3.       p-Acetanisole is commercially available. Since it is one of the abundant components, it should also be screened for cytotoxic activity.

4.       From Table 1, the DPPH radical-scavenging activity has IC50 > 500 ppm. How effective is this? How does it compare with other IC50 values for essential oils?

5.       The component percentages add up to more than 100%. In particular, the value for p-acetanisole (100%) cannot be correct.

6.       What is the effect of oregano essential oil on "normal" cells? Is is selective for tumor cells?

7.       Based on the results, what dosing would be recommended for breast cancer treatment? How should the OEO be administered? Is this dosing feasible?

Some minor corrections to address:

1.       Line 28 says that dried flowers were used, but lines 95-96 states that flowers and leaves were used.

2.       Line 68: Origanum vulgare L. [italics and capitalization].

3.       Be sure to include a space between a measurement and its units (see, for example, lines 99, 298, 417).

4.       Line 103: N2 [subscript]. Line 125: CO2 [subscript]. Line 253: N2 [subscript]. Line 402: Ca2+ [superscript].

5.       Correct the use of capitalization: oregano (line 114), lysis buffer (lines 143-144), catalase (line 416), oregano (line 500).

6.       Line 179: 3,3-dihexyloxacarbocyanine [correct the spelling].

7.       Line 256: “GC and GC-MS analysis” is misleading. It looks as though only GC-MS was carried out.

8.       Use the abbreviation for liter consistently. “L” is better than “l”.

9.       Line 423: Flow cytometric [insert space].

10.   Table 2, KI values for entries 22, 23, and 24, are too different from the calculated AI values. However, values from Adams are in much better agreement (1400 for β-longipinene, 1405 for italicene, and 1407 for longifolene).

11.   Table 2, entry 25: Germacrene A [correct the spelling].

Author Response

We thank the reviewer for your reviewers for the thoughtful insights that allowed to improve the manuscript.  

  1. Based on the data illustrated in Figure 1A, what are the 24-h and 48-h IC50values?

We thank the reviewer for his comment. We calculated 24-h and 48-h IC50 values reported in Figure 1A. This information was added in the new version of the manuscript in Table 1.

  1. What are the IC50 values for p-cymene, γ-terpinene, and thymoquinone (Figure 6)? It would be helpful to determine IC50 values based on these data for easy comparison with others in the literature.

We thank the reviewer for the comment. In Figure 6 are reported the effects of different doses of p-cymene, γ-terpinene, and thymoquinone alone and in combination. Based on these data we calculated the IC50 values of the compounds that are reported in the Table 4 of the new version of the manuscript. 

  1. p-Acetanisole is commercially available. Since it is one of the abundant components, it should also be screened for cytotoxic activity.

We thank the reviewer for his suggestion. There are no data in literature concerning the cytotoxic effect of p-Acetanisole in tumor cells, unlike other compounds such  as p-cymene, γ-terpinene, and thymoquinone whose effects we have evaluated in our conditions.  For this reason, we decided to not tested the effects of p-Acetanisole in previous experiments. However, to respond to the reviewer’s requests, we evaluated the cytotoxic effects of different doses p-Acetanisole, employed alone and in combination with the other three compounds. These results support our previous conclusion that the effect of OEO is related to a combination of several compounds rather than single compounds. These data have been added in the new version of the manuscript. (see pag from 18 to 21) 

  1. From Table 1, the DPPH radical-scavenging activity has IC50 > 500 ppm. How effective is this? How does it compare with other IC50 values for essential oils?

We used DPPH assay to evaluate the antioxidant potential of EOE at the doses of that are effective in breast cancer cells. These data could justify weather OEO did not increase ROS level in early stages of the treatment.

  1. The component percentages add up to more than 100%. In particular, the value for p-acetanisole (100%) cannot be correct.

We thank the reviewer for the comment. The composition of OEO has been evaluated by GC-MS analysis by comparing the content of all the compounds to that present at higher concentration (p-acetanisole) that was considered equal to 100.

  1. What is the effect of oregano essential oil on "normal" cells? Is is selective for tumor cells?

In accordance with the reviewer’s request we tested the effects of OEO in 16HBE14o, a non-tumor human bronchial epithelial cell line, the only normal cell line currently present in our laboratory. We demonstrated that OEO exert a smaller cytotoxic effect in normal than in tumor cells. These data have been added in the new version of the manuscript as supplementary file.

  1. Based on the results, what dosing would be recommended for breast cancer treatment? How should the OEO be administered? Is this dosing feasible?

Our research represents a preliminary study to evaluate the effect of OEO in breast cancer cells in vitro. In vivo studies will be performed to ascertain the doses of OEO that could be recommended for breast cancer treatment. OEO could be administrated by inhalation, with lipid nanoparticles, with microcapsules as reported by other author (Chura SSD, et al. Red sacaca essential oil-loaded nanostructured lipid carriers optimized by factorial design: cytotoxicity and cellular reactive oxygen species levels. Front Pharmacol. 2023 Oct 11;14:1176629. doi: 10.3389/fphar.2023.1176629; Boehm K, et al. Aromatherapy as an adjuvant treatment in cancer care--a descriptive systematic review. Afr J Tradit Complement Altern Med. 2012 Jul 1;9(4):503-18. doi: 10.4314/ajtcam.v9i4.7; Shin SH, et al. Effect of microencapsulated essential oil form Chamaecyparis obtusa on monocyte-derived dendritic cell activation and CD4+ T cell polarization. PLoS One. 2018 Jul 27;13(7):e0201233. doi: 10.1371/journal.pone.0201233)

Some minor corrections to address:

  1. Line 28 says that dried flowers were used, but lines 95-96 states that flowers and leaves were used. Done
  2. Line 68: Origanum vulgareL. [italics and capitalization]. Done line 68
  3. Be sure to include a space between a measurement and its units (see, for example, lines 99, 298, 417). Done
  4. Line 103: N2[subscript]. Line 125: CO2 [subscript]. Line 253: N2 [subscript]. Line 402: Ca2+ [superscript]. Done
  5. Correct the use of capitalization: oregano (line 114), lysis buffer (lines 143-144), catalase (line 416), oregano (line 500). Done
  6. Line 179: 3,3-dihexyloxacarbocyanine [correct the spelling]. Done
  7. Line 256: “GC and GC-MS analysis” is misleading. It looks as though only GC-MS was carried out. Done
  8. Use the abbreviation for liter consistently. “L” is better than “l”. Done
  9. Line 423: Flow cytometric [insert space]. Done
  10. Table 2, KI values for entries 22, 23, and 24, are too different from the calculated AI values. However, values from Adams are in much better agreement (1400 for β-longipinene, 1405 for italicene, and 1407 for longifolene). Thank you for observations.
  11. Table 2, entry 25: Germacrene A [correct the spelling]. Done

Reviewer 2 Report

Comments and Suggestions for Authors

I have read with great interest the manuscript "Cytotoxic effect induced by Sicilian Oregano Essential Oil in Human Breast Cancer Cells" authored by Di Liberto et al. The results obtained by the authors are remarkable and could have translational applications in the field of breast cancer treatment, mainly in combination with other therapeutic interventions such as chemotherapy. In the attached PDF I have made numerous notes, comments, and suggestions that the authors will have to take into account.

Author Response

We thank the reviewer for appreciating the paper. We have modified the manuscript according to your requests. We upload the PDF with the answers, which accompany the revised manuscript.

Round 2

Reviewer 1 Report

Comments and Suggestions for Authors

There are some minor suggestions to consider:

  1. The component percentages add up to more than 100%. In particular, the value for p-acetanisole (100%) cannot be correct.

We thank the reviewer for the comment. The composition of OEO has been evaluated by GC-MS analysis by comparing the content of all the compounds to that present at higher concentration (p-acetanisole) that was considered equal to 100.

This is an unconventional way of presenting concentrations. Please present percentages as percent of total.

  1. What is the effect of oregano essential oil on "normal" cells? Is is selective for tumor cells?

In accordance with the reviewer’s request we tested the effects of OEO in 16HBE14o, a non-tumor human bronchial epithelial cell line, the only normal cell line currently present in our laboratory. We demonstrated that OEO exert a smaller cytotoxic effect in normal than in tumor cells. These data have been added in the new version of the manuscript as supplementary file.

Be sure to add the “Supplementary Materials” availability statement after the Conclusions section.

Table 2, KI values for entries 22, 23, and 24, are too different from the calculated AI values. However, values from Adams are in much better agreement (1400 for β-longipinene, 1405 for italicene, and 1407 for longifolene). Thank you for observations.

Please make these changes in Table 3

Author Response

This is an unconventional way of presenting concentrations. Please present percentages as percent of total.

We thank the reviewer for the comment. We modified table 3.

Be sure to add the “Supplementary Materials” availability statement after the Conclusions section.

Thank you for your suggestion. We inserted the note after the conclusions.

Table 2, KI values for entries 22, 23, and 24, are too different from the calculated AI values. However, values from Adams are in much better agreement (1400 for β-longipinene, 1405 for italicene, and 1407 for longifolene). Thank you for observations.

Please make these changes in Table 3

Thanks to the reviewer for the note. We have modified Table 3 according to the suggested indications.